# Channel Modeling and Quantization Design for 3D NAND Flash Memory

**DOI:** 10.3390/e25070965

**Published:** 2023-06-21

**Authors:** Cheng Wang, Zhen Mei, Jun Li, Feng Shu, Xuan He, Lingjun Kong

**Affiliations:** 1School of Electronic and Optical Engineering, Nanjing University of Science and Technology, Nanjing 210094, China; cheng.wang@njust.edu.cn; 2National Mobile Communications Research Laboratory, Southeast University, Nanjing 210096, China; 3School of Information and Communication Engineering, Hainan University, Haikou 570228, China; shufeng@njust.edu.cn; 4School of Information Science and Technology, Southwest Jiaotong University, Chengdu 611756, China; xhe@swjtu.edu.cn; 5Jinling Institute of Technology, Nanjing 211169, China; kong@jit.edu.cn

**Keywords:** 3D flash memory, information theory, channel modeling, read-voltage thresholds, quantization, LDPC codes

## Abstract

As the technology scales down, two-dimensional (2D) NAND flash memory has reached its bottleneck. Three-dimensional (3D) NAND flash memory was proposed to further increase the storage capacity by vertically stacking multiple layers. However, the new architecture of 3D flash memory leads to new sources of errors, which severely affects the reliability of the system. In this paper, for the first time, we derive the channel probability density function of 3D NAND flash memory by taking major sources of errors. Based on the derived channel probability density function, the mutual information (MI) for 3D flash memory with multiple layers is derived and used as a metric to design the quantization. Specifically, we propose a dynamic programming algorithm to jointly optimize read-voltage thresholds for all layers by maximizing the MI (MMI). To further reduce the complexity, we develop an MI derivative (MID)-based method to obtain read-voltage thresholds for hard-decision decoding (HDD) of error correction codes (ECCs). Simulation results show that the performance with jointly optimized read-voltage thresholds can closely approach that with read-voltage thresholds optimized for each layer, with much less read latency. Moreover, the MID-based MMI quantizer almost achieves the optimal performance for HDD of ECCs.

## 1. Introduction

NAND flash memory has been widely used in consumer electronic devices over the past decade due to non-volatility, fast write and read speed, and low power consumption [1,2]. Moreover, as the development of process technology and multi-level-cell (MLC) technique, the storage capacity and density of NAND flash memory have been greatly increased. However, as the technology continues to scale down, the storage capacity of NAND flash memory has reached its bottleneck and the reliability of flash memory is severely affected by various types of errors, such as program/erase (P/E) cycling errors, cell-to-cell interference, retention interference, and program errors [3]. The reliability of flash memory is not only affected by P/E cycles and data retention time, but also influenced by the temperature and process variations [4,5,6,7].

Recently, three-dimensional (3D) stacking technology has been applied in NAND flash memory, in which several layers are stacked in the vertical direction [8]. With 3D stacking technology, the density and storage capacity of NAND flash memory are tremendously increased compared with two-dimensional (2D) flash memory. However, the new architecture of 3D NAND flash memory leads to new sources of errors [9,10,11]. For example, due to process variations, the memory cells in different layers of 3D NAND flash memory have shown different error characteristics [6,7,12,13]. In addition, early retention loss occurs in the several hours after programming due to fast charge leakage [14]. As a result, the noise characteristics of 3D flash memory channel are significantly different from that of the 2D flash memory and it is worth investigating the channel modeling of 3D flash memory.

To improve the reliability of 3D flash memory system, advanced error correcting codes (ECCs) are essential. Recently, low-density parity-check (LDPC) codes [15] with hard and soft decision decoding have been considered for 3D flash memory [16,17,18]. For LDPC codes with hard-decision decoding (HDD), the performance mainly depends on the raw bit error rate of the flash memory channel [19,20]. The raw bit error rate of 3D flash memory increases when data retention time and P/E cycles increase. To further improve the error-correction performance, LDPC codes with soft-decision decoding can be employed and its performance heavily depends on the accuracy of channel log-likelihood ratios. Therefore, increasing the number of read-voltage thresholds (i.e., quantization levels) will effectively improve the accuracy of log-likelihood ratios [18,21,22]. However, for flash memory channel, high-precision quantization is not feasible due to the read latency restriction. It is essential to design a quantization scheme with limited read-voltage thresholds to achieve near-optimal performance.

### 1.1. Related Work

Many studies have examined the quantization design for 2D flash memory under limited number of read-voltage levels [2,18,21,23,24,25]. These prior works optimized the read-voltage thresholds with perfect knowledge of channel information, e.g., the number of P/E cycles and data retention time. Based on the well-established channel model, they adopted a non-uniform quantization strategy to reduce the number of read-voltage thresholds and improve the performance by using the maximizing the mutual information (MMI)-based, entropy-based, and finite blocklength-based quantization design methods, respectively.

To improve the system reliability for 3D flash memory, several studies have focused on dealing with new error sources (e.g., layer-to-layer variation) in 3D flash memory. For example, a sentinel-cell approach was proposed in [26] to utilize the error characteristics of memory cells for different layers to obtain optimal read-voltage thresholds. A layer variation-aware reading method was proposed to obtain optimal read-voltage thresholds by recording the voltage offset of each layer in a table in flash memory controller [9]. By exploiting asymmetric error characteristics of 3D flash memory, three asymmetric sensing schemes were proposed to reduce the number of sensing levels and maintain considerable performance [27]. A polynomial-based method was proposed to fit the voltage distribution of flash memory, and optimized read-voltage thresholds by a least square method [28]. A process-variation-aware strategy was proposed to reduce uncorrectable bit errors by storing important information in the flash memory block with higher reliability grades [6]. In addition, a voltage compensation strategy based on the reliability grades was proposed to tune the range of sensing voltages for unreliable pages [7]. However, due to the lack of an accurate 3D flash memory channel model, the above work only focused on proposing new strategies to mitigate the adverse effects of specific error sources. It is difficult to systematically design the read-voltage thresholds and analyze the performance theoretically.

In terms of channel modeling and error analysis for 3D flash memory, previous studies in [11,29] analyzed the common error sources in 3D flash memory, such as incremental step pulse programming noise, cell-to-cell interference, and data retention noise. They found new error sources in 3D flash memory (e.g., layer-to-layer variation and early retention loss) and modeled 3D flash memory channel by using Gaussian distribution. In addition, a neural network-based model was proposed to predict the voltage distribution [30]. However, these work failed to derive the joint probability distributions of different types of errors in 3D flash memory.

Therefore, it is essential to analyze the statistics of major errors of 3D flash memory, and derive a mathematical formulation for the channel model. Using this channel model, the mutual information (MI) of 3D flash memory can be derived, and the quantization can be designed theoretically.

### 1.2. Contributions

The main contributions of this paper are as follows:For the first time, we derive an analytic channel model for 3D NAND MLC flash memory, which considers major sources of errors.Rather than optimizing read-voltage thresholds for each layer, we jointly design the quantization for multiple layers of 3D MLC flash memory by MMI. By doing this, the number of total read thresholds for multiple layers can be greatly reduced, such that the storage cost and read latency can be significantly decreased.A dynamic programming (DP) algorithm is proposed to optimize read-voltage thresholds for 3D MLC flash memory by MMI of the joint channel.To further reduce the complexity of DP algorithm, the 3D MLC flash memory channel is simplified and a MI derivative (MID)-based method is developed to obtain read-voltage thresholds for ECCs with HDD.

The remainder of this paper is organized as follows. In Section 2, we derive the channel model for 3D MLC flash memory and further propose a quantized channel model. In Section 3, we formulate the read-voltage thresholds optimization problem of 3D MLC flash memory with multiple layers and utilize MMI-DP algorithm to solve this optimization problem. Then, to further reduce the complexity, we simplify 3D MLC flash memory channel and propose a MID-based quantization scheme. The simulation results are presented in Section 4, and we conclude this paper in Section 5.

## 2. Channel Modeling of 3D NAND MLC Flash Memory

### 2.1. 3D NAND Flash Memory Basics

The 3D NAND flash memory has a vertically stacked structure, which includes word-lines and bit lines located in different layers [8,31]. In 3D NAND flash memory, the information is stored as charges in memory cells [8]. As shown in Figure 1, for MLC flash memory, a memory cell can store two bits of information, which can be represented by four storage states (i.e., $S=\{s_{0},s_{1},s_{2},s_{3}\}$). To write data into a storage cell, the flash memory controller must erase the entire block that this cell belongs to [32]. After erasing, the voltage of this cell is programmed to a specific value by an incremental step pulse programming technique [33]. To read data from a storage cell, the flash memory controller applies read threshold voltage $V_{\mathrm{th}}$ (e.g., $d_{1},d_{2},d_{3}$) to the word-line. By comparing the voltage of storage cell with these read-voltage thresholds, we can determine the stored bits or obtain soft information for further ECC decoding [34].

### 2.2. Channel Modeling

To investigate the optimal read-voltage thresholds design, the channel modeling of 3D NAND MLC flash memory is essential. In this subsection, we propose a simplified channel model for 3D NAND MLC flash memory based on the experimental data in [9] by considering major sources of errors.

#### 2.2.1. Initial Threshold Voltage Distribution

With the reference to [9], the threshold voltage distribution of the memory cell at erased state $s_{0}$ is approximately modeled as a Gaussian distribution $p_{u_{0}}\left(v\right)∼N\left(V_{s_{0}},\sigma_{u}^{2}\right)$ with mean $V_{s_{0}}$ and standard deviation $\sigma_{u}$, respectively. Moreover, the voltages at programmed states (i.e., $s_{1},s_{2},s_{3}$) are generated by the incremental step pulse programming technique. Hence, the threshold voltage distribution of each programmed state $s_{i}$, i=1,2,3, follows a uniform distribution
(1)$$p_{u_{i}}\left(v\right)=\left\{\begin{array}{cc} 1/V_{p}, & v\in \left[V_{s_{i}},V_{s_{i}}+V_{p}\right)\\ 0, & \mathrm{otherwise}, \end{array}\right.$$
where $V_{p}$ denotes the programming step voltage and $V_{s_{i}}$ denotes the target programmed voltage of programmed state $s_{i}$. The programmed cells are also affected by the programming noise, which can also be modeled by a Gaussian distribution $n_{p_{i}}\left(v\right)∼N\left(0,\sigma_{p_{i}}^{2}\right)$, i=1,2,3.

#### 2.2.2. Program/Erase Cycling Errors

P/E cycling errors $n_{\mathrm{pe}}$ appear immediately after the programming and erasing operations. As the number of P/E cycles increases, more and more electrons are trapped in the transistor, which will reduce the tunneling efficiency and result in inaccurate charge transport. The P/E cycling errors can be approximately modeled as a Gaussian distribution $p_{n_{\mathrm{pe}}}\left(v\right)∼N\left(\mu_{\mathrm{pe}},\sigma_{\mathrm{pe}}^{2}\right)$, where $\mu_{\mathrm{pe}}$ and $\sigma_{\mathrm{pe}}$ are the mean and standard deviation, respectively, [9].

#### 2.2.3. Cell-to-Cell Interference

Due to the parasitic capacitance-coupling effect, one cell can affect the voltage distribution of its neighbor cells. The voltage of the victim cell is affected by that of it adjacent cells, leading to program interference errors. Note that, different from 2D MLC flash memory, the voltage of the victim cell of 3D MLC flash memory can also be affected by its neighbor layers. As mentioned in [29,35], the cell-to-cell interference can be estimated and mitigated by the pre-distortion or post-compensation techniques.

#### 2.2.4. Early Retention Loss

As the data retention time increases, the charge loss will move the voltage distribution of flash memory to lower states [36]. Compared with 2D flash memory, the memory cells in 3D flash memory suffer from early retention loss, which presents a large amount of charge loss within a few minutes after being programmed. According to [9,11], the early retention noise can be modeled as a Gaussian distribution, $p_{n_{r}}\left(v\right)∼N\left(\mu_{r},\sigma_{r}^{2}\right)$, where $\mu_{r}$ and $\sigma_{r}$ are the mean and standard deviation, respectively.

#### 2.2.5. Layer-to-Layer Process Variation

Due to process variation, the error characteristics of each layer are quite different. The layer-to-layer process variation of each layer can be modeled as a Gaussian distribution $p_{n_{l_{k}}}\left(v\right)∼N\left(\mu_{l_{k}},\sigma_{l_{k}}^{2}\right)$[9], where $\mu_{l_{k}}$ and $\sigma_{l_{k}}$ are the mean and standard deviation of the data stored in the *k*-th layer, respectively. According to the experimental results in [9], it shows the mean and standard deviation of each state vary with the layer. To evaluate the influence of process variations for different layers, we set the first layer as the reference layer and fit the mean and standard deviation of other layers using a polynomial fitting method.

Finally, the overall voltage distribution can be calculated by the convolution integral of initial voltage distribution functions with other major noises [2,21]. As mentioned above, the erased state $s_{0}$ is not affected by programming noise, its voltage distribution for the *k*-th layer is given as
(2)$$\begin{array}{c} \begin{array}{cc} p_{s_{0,k}}\left(v\right) & =p_{u_{0}}⊗p_{n_{\mathrm{pe}}}⊗p_{n_{r}}⊗p_{n_{l_{k}}}\\ & =\frac{1}{\sigma_{s_{0,k}}\sqrt{2\pi }}e^{-\frac{{\left(v-\mu_{s_{0,k}}\right)}^{2}}{2\sigma_{s_{0,k}}^{2}}}, \end{array} \end{array}$$
where ⊗ denotes the convolution operation, $\mu_{s_{0,k}}=V_{s_{0}}+\mu_{\mathrm{pe}}+\mu_{r}+\mu_{l_{k}}$, $\sigma_{s_{0,k}}^{2}=\sigma_{u}^{2}+\sigma_{\mathrm{pe}}^{2}+\sigma_{r}^{2}+\sigma_{l_{k}}^{2}$. The voltage distributions of the programmed states $s_{i}$ (i=1,2,3) for the *k*-th layer are given as
(3)$$\begin{array}{c} \begin{array}{cc} & p_{s_{i,k}}\left(v\right)=p_{u_{i}}⊗n_{p_{i}}⊗p_{n_{\mathrm{pe}}}⊗p_{n_{r}}⊗p_{n_{l_{k}}}\\ & =\frac{1}{2V_{p}}\left(\mathrm{erf}\left(\frac{V_{s_{i}}+V_{p}-v-\mu_{s_{i,k}}}{\sqrt{2}\sigma_{s_{i,k}}}\right)\right)-\\ & \frac{1}{2V_{p}}\left(\mathrm{erf}\left(\frac{V_{s_{i}}-v-\mu_{s_{i,k}}}{\sqrt{2}\sigma_{s_{i,k}}}\right)\right),\;i=1,2,3, \end{array} \end{array}$$
where *i* denotes the index of the storage states, $\mu_{s_{i,k}}=\mu_{\mathrm{pe}}+\mu_{r}+\mu_{l_{k}}$, $\sigma_{s_{i,k}}^{2}=\sigma_{p_{i}}^{2}+\sigma_{\mathrm{pe}}^{2}+\sigma_{r}^{2}+\sigma_{l_{k}}^{2}$, and erf(*) denotes the Gauss error function.

According to the experimental data of 3D NAND flash memory given in [9], the parameters of voltage distributions with layer-to-layer variation can be obtained by fitting the experimental data using the least square method, listed in Table 1. Similar to [9], the voltage values are normalized in this work. Finally, the parameters of 3D flash memory voltage distribution can be calculated by $\mu_{s_{i,k}}=\mu_{s_{i}}^{*}+\mu_{l_{i,k}}$, and $\sigma_{s_{i,k}}^{2}={\left(\sigma_{s_{i}}^{*}\right)}^{2}+\sigma_{l_{i,k}}^{2}$.

### 2.3. Quantized Model of 3D NAND Flash Memory

In this paper, we assume that the 3D NAND MLC flash memory has a total of *K* layers. Since the read process transforms voltages into discrete values, the flash memory channel is quantized into a discrete memoryless channel. For MLC flash memory, each cell can store two bits of information and has four storage states, $s_{0},s_{1},s_{2}$, and $s_{3}$. Figure 1 shows an example of the voltage distribution with read-voltage thresholds (i.e., red dotted line). Let $S=\{s_{0},s_{1},s_{2},s_{3}\}$ denote the storage states of MLC flash memory. Given *J* thresholds, the flash memory channel is quantized into *J+1* levels. Let $D_{k}=\left\{d_{1,k},d_{2,k},\ldots ,d_{J,k}\right\}$ denote *J* read-voltage levels, and $R_{k}=\left\{r_{0,k},r_{1,k},\ldots ,r_{J,k}\right\}$ denote *J+1* channel outputs in the *k*-th layer, where $r_{j,k}\in \left[d_{j,k},d_{j+1,k}\right)$ with $d_{0,k}=-\infty$ and $d_{J+1,k}=+\infty$. In addition, $d_{1,k}<d_{2,k}<\cdots <d_{J,k}$. As shown in Figure 2, in the *k*-th layer, read voltage quantization produces a discrete memoryless channel with inputs $\left\{s_{0},s_{1},s_{2},s_{3}\right\}$ and outputs $\left\{r_{0,k},r_{1,k},\cdots ,r_{J,k}\right\}$. The transition probability of storing $s_{i}$ and quantizing as $r_{j,k}$ in the *k*-th layer is given by
(4)$$\begin{array}{c} \begin{array}{c} P\left(r_{j,k}|s_{i}\right)=\int_{d_{j,k}}^{d_{j+1,k}}p_{s_{i,k}}\left(v\right)dv, \end{array} \end{array}$$
and
(5)$$\begin{array}{c} \begin{array}{c} P\left(r_{j,k}\right)=\sum_{i=0}^{3}P\left(s_{i}\right)P\left(r_{j,k}|s_{i}\right), \end{array} \end{array}$$
where $P\left(s_{i}\right)$ is the probability of the flash memory cell stores $s_{i}$. Due to the fact that the scrambler in the flash controller randomly perturbs data bits to ensure the stored data bits (1 or 0) are evenly distributed [20], the probability of input is $P\left(s_{i}\right)=\frac{1}{4}$. With this quantized model of 3D flash memory, the MI of the 3D MLC flash memory channel will be derived in the next section, and the read-voltage thresholds can be optimized by MMI.

## 3. Quantization Design for 3D NAND Flash Memory

In this section, the MI of 3D NAND MLC flash memory channel is derived based on the quantized channel model. Then, a MMI-DP algorithm is proposed to optimize read-voltage thresholds. Moreover, to reduce the complexity of DP-based MMI quantization algorithm, we propose a MID-based quantization read-voltage thresholds design for ECCs with HDD.

### 3.1. MI for 3D NAND MLC Flash Memory

The voltage distributions of different layers in 3D NAND MLC flash memory have different statistical properties. Conventional methods, such as the MMI-based quantization [18], entropy-based quantization [21], can still optimize read-voltage thresholds for each layer in 3D NAND flash memory. However, applying different read-voltage thresholds in different layers requires more read operations. Since the read operation is a time-consuming process in flash memory [37], the conventional methods of optimizing read-voltage thresholds for each layer are not desired for 3D NAND flash memory. Therefore, it is necessary to jointly optimize read-voltage thresholds for all layers to reduce the read overhead of 3D flash memory.

For 3D NAND flash memory channel with *K* layers, let S and R denote the inputs and outputs of this channel, respectively. Then, the MI of 3D NAND flash memory channel can be calculated by
(6)$$\begin{array}{c} \begin{array}{cc} I(S;R) & =I(S_{1}S_{2}\cdots S_{K};R_{1}R_{2}\cdots R_{K})\\ & =H\left(R\right)-H\left(R|S\right)=H\left(R_{1}R_{2}\cdots R_{K}\right)-\sum_{k=1}^{K}H\left(R_{k}|S_{k}\right), \end{array} \end{array}$$
where $S_{k}\in S$ and $R_{k}\in R$, H(*) denotes the entropy.

The MI for the *k*-th layer 3D NAND flash memory channel can be calculated by Equation (Equation 7).
(7)$$\begin{array}{cc} I\left(S_{k};R_{k}\right) & =H\left(R_{k}\right)-H\left(R_{k}|S_{k}\right)=H\left(S_{k}\right)-H\left(S_{k}|R_{k}\right)\\ \; & =\sum_{j=0}^{J}\sum_{i=0}^{3}P\left(s_{i}\right)P\left(r_{j,k}|s_{i}\right)\log\frac{P\left(r_{j,k}|s_{i}\right)}{\sum_{i=1}^{3}P\left(r_{j,k}|s_{i}\right)P\left(s_{i}\right)}\\ \; & =\sum_{j=0}^{J}\sum_{i=0}^{3}P(s_{i})\int_{d_{j,k}}^{d_{j+1,k}}p_{s_{i,k}}\left(v\right)dvs.\log\frac{\int_{d_{j,k}}^{d_{j+1,k}}p_{s_{i,k}}\left(v\right)dv}{\sum_{i=0}^{3}P(s_{i})\int_{d_{j,k}}^{d_{j+1,k}}p_{s_{i,k}}\left(v\right)dv},\;k=1,2,\cdots ,K. \end{array}$$

The sum of MI for all layers is given by
(8)$$\begin{array}{c} \begin{array}{cc} & \sum_{k=1}^{K}I\left(S_{k};R_{k}\right)=\sum_{k=1}^{K}\left[H\left(R_{k}\right)-H\left(R_{k}|S_{k}\right)\right]\\ & =\sum_{k=1}^{K}\left[H\left(R_{k}\right)\right]-\sum_{k=1}^{K}\left[H\left(R_{k}|S_{k}\right)\right]. \end{array} \end{array}$$

The difference between Equations (Equation 6) and (Equation 8) is
(9)$$\begin{array}{c} \begin{array}{c} I\left(S;R\right)-\sum_{k=1}^{K}I\left(S_{k};R_{k}\right)=H\left(R\right)-\sum_{k=1}^{K}H\left(R_{k}\right). \end{array} \end{array}$$

According to the definition of entropy, we have
(10)$$\begin{array}{c} \begin{array}{c} H\left(R\right)=H\left(R_{1}R_{2}\cdots R_{K}\right)\leq \sum_{k=1}^{K}H\left(R_{k}\right). \end{array} \end{array}$$

As a result,
(11)$$\begin{array}{c} \begin{array}{c} I\left(S;R\right)\leq \sum_{k=1}^{K}I\left(S_{k};R_{k}\right). \end{array} \end{array}$$

For 3D NAND flash memory, the channel of each layer is a discrete memoryless channel. However, these channels are correlated due to the presence of the cell-to-cell interference. As mentioned in [11,35], the cell-to-cell interference can be mitigated by pre-distortion/post-compensation technique. Therefore, we assume that the channel of each layer is independent of each other. Hence, the equality sign of Equation (Equation 11) holds [38], yields
(12)$$\begin{array}{c} \begin{array}{c} I\left(S;R\right)=\sum_{k=1}^{K}I\left(S_{k};R_{k}\right). \end{array} \end{array}$$

The optimal read-voltage thresholds can be derived by MMI in Equation (Equation 12). To jointly optimize the read-voltage thresholds for all layers, the optimization problem can be formulated as
(13)$$\begin{array}{c} \begin{array}{cc} & P:\;\max\;\sum_{k=1}^{K}I\left(S_{k};R_{k}\right)\\ & s.t.\;\;0<d_{1}<d_{2}<\cdots <d_{J}<\infty . \end{array} \end{array}$$

Similarly, to optimize the read-voltage thresholds for each layer, this optimization problem can be formulated as
(14)$$\begin{array}{c} \begin{array}{cc} & P:\;\max\;I\left(S_{k};R_{k}\right)\\ & s.t.\;\;0<d_{1,k}<d_{2,k}<\cdots <d_{J,k}<\infty ,\\ & \;\;\;k=1,2,\cdots ,K. \end{array} \end{array}$$

### 3.2. MMI Quantization Design

DP is an effective method to solve the optimization of multi-stage decision-making process. Specifically, the problems in Equations (Equation 13) and (Equation 14) can be decomposed into several smaller local problems, and then be solved in turn according to the recurrence relationship of the local problems to reach the global optimum [39]. In this subsection, we propose a MMI-DP-based algorithm in Algorithm 1 to optimize the read-voltage thresholds for 3D MLC flash memory channel.

First, the flash memory channel is uniformly quantized into *N* intervals and the boundaries are $\left\{a_{0},a_{1},\cdots ,a_{N}\right\}$, where J≪N (e.g., N=1000). We set $a_{0}=-\infty$, $a_{1}=V_{s_{0}}-5\times \sigma_{s_{0}}$, $a_{N-1}=V_{s_{3}}+5\times \sigma_{s_{3}}$, and $a_{N}=\infty$. Then, other boundaries can be obtained by $a_{n}=a_{1}+\left(n-1\right)\left(a_{N-1}-a_{1}\right)/\left(N-2\right)$. Second, the MI in Equation (Equation 7) can also be written as $I\left(S_{k};R_{k}\right)=H\left(S_{k}\right)-H\left(S_{k}|R_{k}\right)$, where $H(S_{k})$ is a constant. Therefore, the maximization of $I(S_{k};R_{k})$ is equivalent to minimizing $H(S_{k}|R_{k})$. Hence, the optimization problems Equations (Equation 13) and (Equation 14) can be reformulated as
(15)$$\begin{array}{c} \begin{array}{cc} & P:\;\min\;\sum_{k=k_{0}}^{k_{1}}H\left(S_{k}|R_{k}\right)\\ & s.t.\;\;0<d_{1}<d_{2}<\cdots <d_{J}<\infty , \end{array} \end{array}$$
where $k_{0}$ and $k_{1}$ denote the layer index of 3D flash memory, respectively. To jointly optimized thresholds for all layers, we set $k_{0}=1$, $k_{1}=K$. To optimize thresholds for each layer, we set $k_{0}=k_{i}$, $k_{1}=k_{i}$, where $k_{i}$ denotes the target layer index. The optimal read-voltage thresholds can be obtained by: (16)$$\begin{array}{c} \begin{array}{c} D^{*}=\left\{d_{1}^{*},\cdots ,d_{J}^{*}\right\}=\underset{\left\{d_{1},\cdots ,d_{J}\right\}\subset \left\{a_{0},\cdots ,a_{N}\right\}}{\arg\min}\sum_{k=k_{0}}^{k_{1}}H\left(S_{k}|R_{k}\right). \end{array} \end{array}$$

The conditional entropy of *k*-th layer $H\left(S_{k}|R_{k}\right)$ can be calculated by
(17)$$\begin{array}{c} \begin{array}{c} H\left(S_{k}|R_{k}\right)=\sum_{j=0}^{J}\Delta \left(d_{j},d_{j+1}\right), \end{array} \end{array}$$
where
(18)$$\begin{array}{c} \begin{array}{c} \Delta \left(d_{j},d_{j+1}\right)=\sum_{i=0}^{3}P\left(s_{i}\right)P\left(r_{j,k}|s_{i}\right)\log\frac{\sum_{i=0}^{3}P\left(r_{j,k}|s_{i}\right)P\left(s_{i}\right)}{P\left(r_{j,k}|s_{i}\right)P\left(s_{i}\right)}. \end{array} \end{array}$$

The MMI quantizer is a sequential deterministic quantizer [39]. We use C(N,J+1) to denote the cost function of quantizing $\left\{a_{0},a_{1},\cdots ,a_{N}\right\}$ into J+1 levels. Let $C^{*}\left(N,J+1\right)$ denote the cost function of the optimal solution in Equation (Equation 15). For a sequential deterministic quantizer, it can be decomposed into several smaller quantizers. Then, we have
(19)$$\begin{array}{c} \begin{array}{cc} & C^{*}\left(N,J+1\right)=\sum_{k=k_{0}}^{k_{1}}\sum_{j=0}^{J}\Delta \left(d_{j}^{*},d_{j+1}^{*}\right)\\ & =C^{*}\left(\lambda_{J}^{*},J\right)+\sum_{k=k_{0}}^{k_{1}}\Delta \left(a_{\lambda_{J}^{*}},a_{N}\right)\\ & =\min_{J\leq \lambda_{J}\leq N}C^{*}\left(\lambda_{J},J\right)+\sum_{k=k_{0}}^{k_{1}}\Delta \left(a_{\lambda_{J}},a_{N}\right), \end{array} \end{array}$$
where $\left\{\lambda_{1}^{*},\lambda_{2}^{*},\cdots ,\lambda_{J}^{*}\right\}\subset \left\{1,2,\cdots ,N-1\right\}$ and $\left\{a_{\lambda_{1}^{*}},a_{\lambda_{2}^{*}},\cdots ,a_{\lambda_{J}^{*}}\right\}$ are the optimal solutions of each smaller quantizer. From Equation (Equation 19), the optimal solution $D^{*}$ can be obtained by solving sub-problems in a recursive manner. The complexity of solving these problems by Algorithm 1 is $O({\left(N-J\right)}^{2}J)$ [39].
**Algorithm 1:** MMI-DP algorithm for searching optimal read-voltage thresholds in 3D flash memory.**Input:***J*, *N*, *K*, $a_{1}$, $a_{N-1}$, $k_{0}$, $k_{1}$.**Output:**$D^{*}=\left\{d_{1}^{*},\cdots ,d_{J}^{*}\right\}$.
1:**for**n=2; n<N; n++**do**2:    $a_{n}=a_{1}+\left(n-1\right)\left(a_{N-1}-a_{1}\right)/\left(N-2\right)$;3:**end for**4:$a_{0}=-\infty$, $a_{N}=\infty$;5:**for**n=1; n≤N; n++**do**6:    $C\left(n,1\right)=\sum_{k=k_{0}}^{k_{1}}\Delta \left(a_{0},a_{n}\right)$;7:**end for**8:**for**q=2; q≤J+1; q++**do**9:    **for** m=q; m≤N−J+q−1; m++ **do**10:        C(m,q)=∞;11:        **for** t=q−1; t<m; t++ **do**12:           **if** $C\left(m,q\right)>C\left(t,q-1\right)+\sum_{k=k_{0}}^{k_{1}}\Delta \left(a_{t},a_{m}\right)$ **then**13:               $C\left(m,q\right)=C\left(t,q-1\right)+\sum_{k=k_{0}}^{k_{1}}\Delta \left(a_{t},a_{m}\right)$;14:               λ(m,q)=t;15:           **end if**16:        **end for**17:    **end for**18:**end for**19:*t* = *N*;20:**for**i=J+1; i>1; i−−**do**21:    t=λ(t,i), $d_{i-1}^{*}=a_{t}$;22:**end for**23:Return $D^{*}=\left\{d_{1}^{*},\cdots ,d_{J}^{*}\right\}$.

To evaluate the performance of our proposed joint quantization design, the performance of the thresholds optimized for each layer by MMI is served as the benchmark. Note that this quantization design method requires a number of read operations which will result in a large read latency.

To examine the performance of Algorithm 1, the MI of 3D flash memory channel with different quantization schemes are presented in Figure 3. Note that the MMI-DP optimized for the first layer is to apply read-voltage thresholds optimized for the first layer to all layers. As illustrated in Figure 3, all MMI-DP based methods can approach the MI of 3D flash memory channel (with quantization level J=∞) by using only nine read-voltage thresholds. Moreover, the MI of our proposed joint optimization algorithm is higher than that optimized for the first layer, and close to the optimum (i.e., optimized for each layer). The impact of layer-to-layer variation on the voltage distribution of $s_{0}$ results in a higher occurrence of errors in the upper layers [9]. The proposed joint optimization algorithm is capable of designing voltage thresholds to effectively mitigate these errors.

### 3.3. Read Thresholds Design for Hard Decision Decoding

In practical applications, to reduce the read latency and power consumption, the HDD of ECCs needs to be performed once before soft-decision decoding. Therefore, it is of great significance to design read-voltage thresholds for HDD of ECCs. Although the proposed DP algorithm can also design read-voltage thresholds for HDD, its computational complexity is high. To solve this problem, we proposed a MID-based quantization scheme to further reduce the complexity.

As described in Section 3. A, by MMI in Equation (Equation 12), we can optimize the read-voltage thresholds for flash memory. Since Equation (Equation 13) is locally concave, we can optimize the thresholds for HDD by calculating the derivative of the MI, given by
(20)$$\begin{array}{c} \begin{array}{c} \frac{dI(S;R)}{dh_{j}}=0, \end{array} \end{array}$$
where j=1,2,3, and $h_{j}$ denotes the read-voltage threshold for HDD of ECCs.
(21)$$\begin{array}{cc} I(S_{k};R_{k}) & =\sum_{j=j-1}^{j}\sum_{i=i-1}^{i}\frac{1}{2}\int_{h_{j}}^{h_{j+1}}p_{s_{i,k}}\left(v\right)dv\log\frac{\int_{h_{j}}^{h_{j+1}}p_{s_{i,k}}\left(v\right)dv}{\sum_{i=i-1}^{i}\frac{1}{2}\int_{h_{j}}^{h_{j+1}}p_{s_{i,k}}\left(v\right)dv}\\ \; & =\frac{1}{2}\int_{-\infty }^{h_{j}}p_{s_{i-1,k}}\left(v\right)dv\log\frac{\int_{-\infty }^{h_{j}}p_{s_{i-1,k}}\left(v\right)dv}{\frac{1}{2}\left(\int_{-\infty }^{h_{j}}p_{s_{i-1,k}}\left(v\right)+p_{s_{i,k}}\left(v\right)dv\right)}+\\ \; & \;\;\frac{1}{2}\int_{h_{j}}^{\infty }p_{s_{i-1,k}}\left(v\right)dv\log\frac{\int_{h_{j}}^{\infty }p_{s_{i-1,k}}\left(v\right)dv}{\frac{1}{2}\left(\int_{h_{j}}^{\infty }p_{s_{i-1,k}}\left(v\right)+p_{s_{i,k}}\left(v\right)dv\right)}+\\ \; & \;\;\frac{1}{2}\int_{-\infty }^{h_{j}}p_{s_{i,k}}\left(v\right)dv\log\frac{\int_{-\infty }^{h_{j}}p_{s_{i,k}}\left(v\right)dv}{\frac{1}{2}\left(\int_{-\infty }^{h_{j}}p_{s_{i-1,k}}\left(v\right)+p_{s_{i,k}}\left(v\right)dv\right)}+\\ \; & \;\;\frac{1}{2}\int_{h_{j}}^{\infty }p_{s_{i,k}}\left(v\right)dv\log\frac{\int_{h_{j}}^{\infty }p_{s_{i,k}}\left(v\right)dv}{\frac{1}{2}\left(\int_{h_{j}}^{\infty }p_{s_{i-1,k}}\left(v\right)+p_{s_{i,k}}\left(v\right)dv\right)},\\ \; & \;\;j=1,2,3,\;i=1,2,3. \end{array}$$

However, it is difficult to derive an analytical solution for Equation (Equation 20). We propose a MID-based method to optimize read-voltage thresholds by dividing the 2-bits quantization into three 1-bit quantization problems. As shown in Figure 4, each 1-bit quantization mainly depends on the adjacent state of flash memory, i.e., the red and blue region. With this simplification, the flash memory channel can be regarded as three binary asymmetric channels and the optimization problem can be solved by finding the root of the following equation: (22)$$\begin{array}{c} \begin{array}{c} \frac{d\sum_{k=k_{0}}^{k_{1}}I\left(S_{k};R_{k}\right)}{dh_{j}}=0,\;\;j=1,2,3. \end{array} \end{array}$$

According to Equation (Equation 7), the MI between two adjacent states in *k*-th layer can be calculated by
(23)$$\begin{array}{c} \begin{array}{c} I\left(S_{k};R_{k}\right)=\sum_{j=j-1}^{j}\sum_{i=i-1}^{i}P\left(s_{i}\right)P\left(r_{j,k}|s_{i}\right)\log\frac{P\left(r_{j,k}|s_{i}\right)}{\sum_{i=i-1}^{i}P\left(r_{j,k}|s_{i}\right)P\left(s_{i}\right)},\;i=1,2,3,\;j=1,2,3, \end{array} \end{array}$$
where $P\left(s_{i}\right)=\frac{1}{2}$.

By expanding Equation (Equation 23) to Equation (Equation 21) and substitute it into Equation (Equation 22), we can calculate the solution of Equation (Equation 22). Let $h_{j}^{*}$ denote the solution of Equation (Equation 22). To determine $h_{j}^{*}$, we first compute the derivative of Equation (Equation 21) with respect to $h_{j}$. The detailed derivative is given in Appendix A. By using the proposed MID-based quantization, we can obtain the solution of Equation (Equation 22) which is a suboptimal solution of Equation (Equation 20). To solve Equation (Equation 22), the root can be found by employing binary search method. Hence, the complexity of the MID-based quantization is O(logn), where *n* is the number of samples, and n=1024 has been found to be sufficient. Therefore, the computational complexity of the MID-based quantization is significantly lower than the MMI-DP algorithm which is $O({\left(N-J\right)}^{2}J)$.

To evaluate the performance of thresholds designed by MID, the symbol error probability (SEP) of 3D MLC flash memory with full channel knowledge is further derived. For given $\left\{h_{1},h_{2},h_{3}\right\}$, the SEP of an uncoded 3D flash memory channel can be calculated by
(24)$$\begin{array}{c} \begin{array}{c} P_{e}=\sum_{k=k_{0}}^{k_{1}}\frac{P_{e}\left(s_{0},k\right)+P_{e}\left(s_{1},k\right)+P_{e}\left(s_{2},k\right)+P_{e}\left(s_{3},k\right)}{4(k_{1}-k_{0}+1)}, \end{array} \end{array}$$
where
(25)$$\begin{array}{c} \begin{array}{cc} & P_{e}\left(s_{0},k\right)=\int_{h_{1}}^{\infty }p_{s_{0,k}}\left(v\right)dv,P_{e}\left(s_{1},k\right)=\int_{-\infty }^{h_{1}}p_{s_{1,k}}\left(v\right)dv+\int_{h_{2}}^{\infty }p_{s_{1,k}}\left(v\right)dv,\\ & P_{e}\left(s_{2},k\right)=\int_{-\infty }^{h_{2}}p_{s_{2,k}}\left(v\right)dv+\int_{h_{3}}^{\infty }p_{s_{2,k}}\left(v\right)dv,P_{e}\left(s_{3},k\right)=\int_{-\infty }^{h_{3}}p_{s_{3,k}}\left(v\right)dv. \end{array} \end{array}$$

To evaluate the performance of our proposed MID scheme, the thresholds optimized by minimizing SEP (MSEP) is served as a benchmark. To find the thresholds that minimize the SEP, a cross iterative searching algorithm is adopted by utilizing genetic and iterative searching algorithm [24]. Through plenty of iterations, the cross iterative searching algorithm can approach the global optimum for given number of quantization levels. Therefore, it can serve as the benchmark of our proposed low-complexity MID-based quantization design.

Figure 5 compares the MI of 3D flash memory channel with the MSEP, MMI-DP, and MID quantizers. The MI of the unquantized channel is also included as a reference. First, as shown in Figure 5, the MI of MMI-DP quantizer and MSEP quantizer are almost the same, which indicates the DP algorithm has reached the global optimum. Second, the MI of the MID quantizer close approaches to that of the MMI-DP quantizer, which demonstrates the effectiveness of the proposed MID quantizer. Finally, the MI of joint optimization scheme for 3D flash memory approaches that of optimizing for each layer in 3D flash memory, which verifies that the joint optimization scheme can achieve near-optimal performance for 3D flash memory.

## 4. Numerical and Simulation Results

In the simulations, the total number of layer in 3D flash memory is set to 30. We first investigate the uncoded SEP performance with different quantization schemes over different P/E cycles at $t=5\times 10^{6}$ s. The quantization scheme that directly minimizes the SEP is served as a benchmark. With read-voltage thresholds optimized by these quantizers, the SEP of 3D flash memory channel can be calculated by Equation (Equation 24). As shown in Figure 6, the SEP of these quantization schemes is consistent with their MI in Figure 5. It is observed that the SEP of the MID quantization is close to that of the MMI-DP quantization. In addition, the quantization scheme of joint optimized for all layers shows superior performance when compared with the first layer optimization quantization scheme.

Next, the LDPC-coded frame error rate (FER) performance of different quantization scheme is examined. The decoding algorithm of LDPC codes is sum-product algorithm with maximum 25 iterations ($I_{\max}=25$). In our simulations, a *4K-code* is employed and constructed by progressive-edge-growth algorithm [21]. The code length for *4K-code* is 4544 bits with code rate 0.9. The degree distribution of this code is given as
(26)$$\begin{array}{c} \begin{array}{cc} & \epsilon \left(x\right)=0.0682x+0.1822x^{2}+0.1329x^{3}+0.6167x^{4},\\ & \theta \left(x\right)=0.22x^{38}+0.78x^{39}, \end{array} \end{array}$$
where ϵ(x) and θ(x) are the variable-node and check-node degree distribution optimized by density-evolution, respectively.

First, the LDPC-coded performance of different quantizers with 3-level quantization is presented in Figure 7. The FER of the MSEP quantizer is also included as the benchmark. Similarly, it can be seen that, the FER performance of joint optimization for all layers by MID scheme is superior to that of optimization for the first layer and close to the optimal performance. For example, at FER = $10^{-4}$, the proposed quantizer improves the endurance of 3D flash memory by 2000 P/E cycles compared with the quantizer that optimized for the first layer.

To enable soft-decision decoding, more read-voltage thresholds are essential. Figure 8 illustrates the FER performance of *4K-code* with six read-voltage thresholds. In addition, the performance of the uniform quantizer is also included as a reference. Note that the uniform quantization has 20 read-voltage thresholds and our designed non-uniform quantizers has 6 read-voltage thresholds. It is observed that the FER performance of joint optimized thresholds is much better than that of optimized for the first layer and close to the optimum (i.e., the thresholds optimized for each layer). The FER performance is consistent with the MI in Figure 3, which all demonstrate the superiority of our proposed algorithm. For example, at FER = $10^{-4}$, the proposed algorithm improves the endurance of 3D flash memory by 3100 P/E cycles compared with that optimized for the first layer. Compared with uniform quantization, the proposed algorithm only needs 6 read-voltage thresholds to surpass the performance of 20 uniform thresholds. This demonstrates that non-uniform read-voltage design reduces the number of read operations while still maintaining desirable error-correction performance.

## 5. Conclusions

In this paper, we have derived the channel model for 3D MLC flash memory based on the experimental data. Next, the 3D MLC flash memory with *K* layers is regarded as a joint channel and the channel capacity has been derived. By maximizing the MI of 3D flash memory channel, we have further proposed a MMI-DP algorithm to optimize read-voltage thresholds. In addition, to reduce the complexity of the MMI-DP algorithm, we have simplified the 3D MLC flash memory channel model and proposed a MID-based quantization scheme to obtain read-voltage thresholds for ECCs with HDD. Simulation results have shown that the FER performance of our proposed joint optimization algorithm can almost achieve the performance that is optimized for each layer with much less read-voltage thresholds, such that the read latency can be significantly reduced.

## Figures and Tables

**Figure 1 entropy-25-00965-f001:**
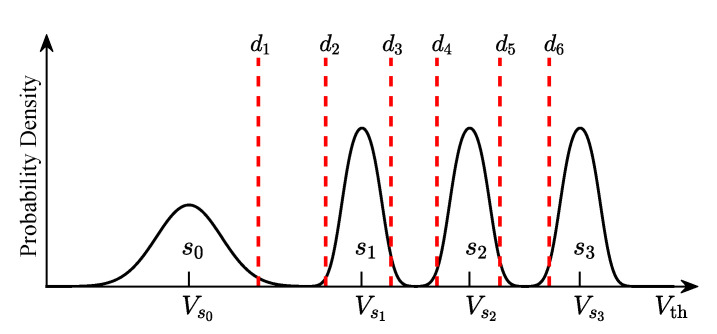
Illustration of voltage distributions and read-voltage thresholds for MLC flash memory.

**Figure 2 entropy-25-00965-f002:**
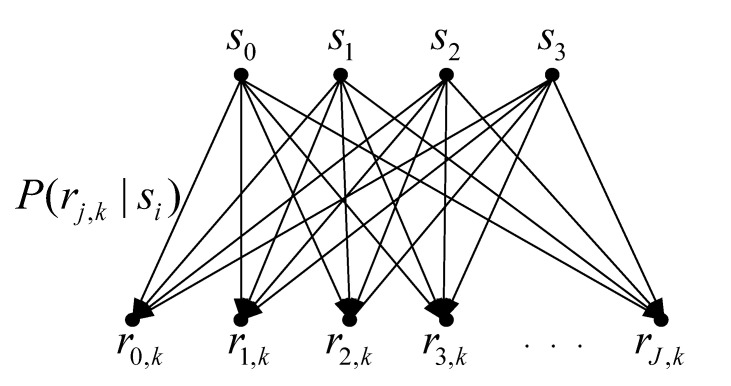
Equivalent discrete memoryless channel for 3D NAND MLC flash memory in the *k*-th layer.

**Figure 3 entropy-25-00965-f003:**
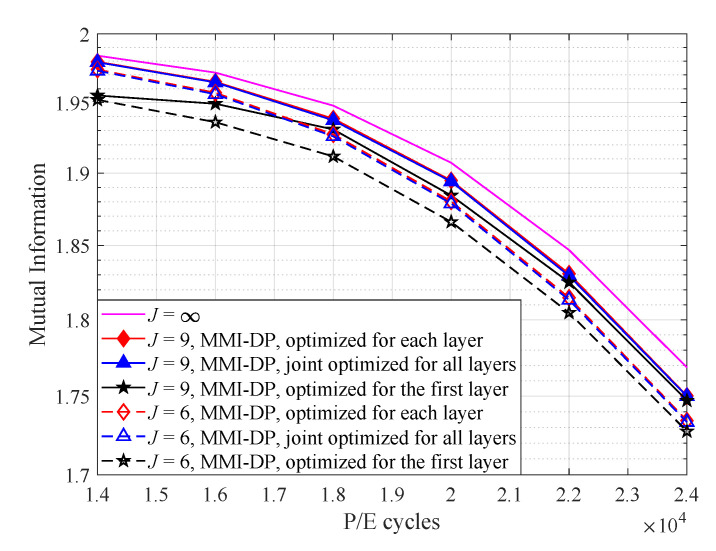
The mutual information of quantized 3D flash memory channel with the MMI quantizer at $t=10^{4}$ s.

**Figure 4 entropy-25-00965-f004:**
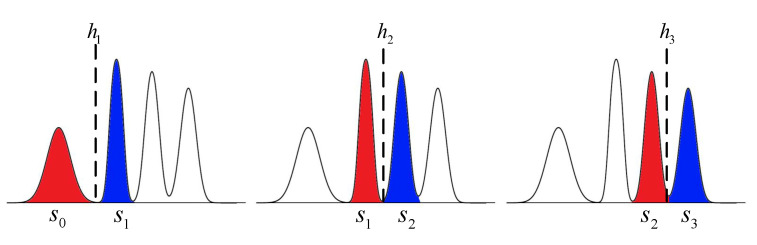
Simplified model of MLC flash memory.

**Figure 5 entropy-25-00965-f005:**
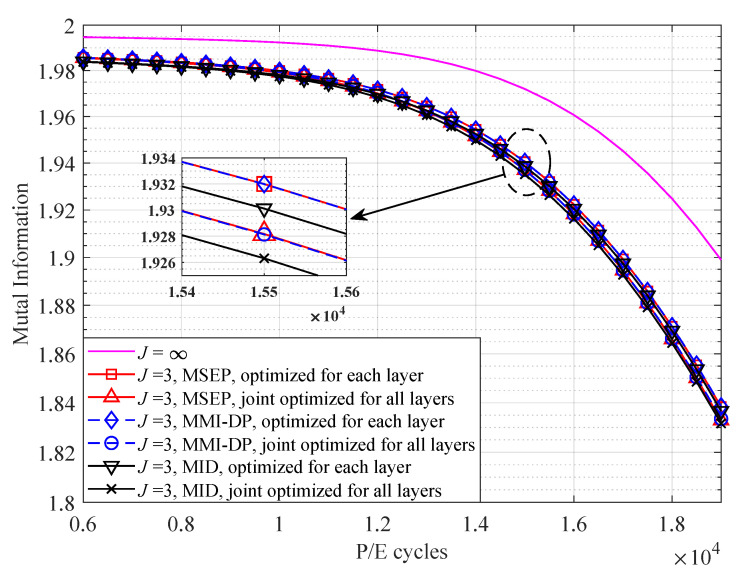
The MI of the MSEP, MMI-DP, and MID quantizers with 3 read-voltage thresholds over different P/E cycles at $t=5\times 10^{6}$ s.

**Figure 6 entropy-25-00965-f006:**
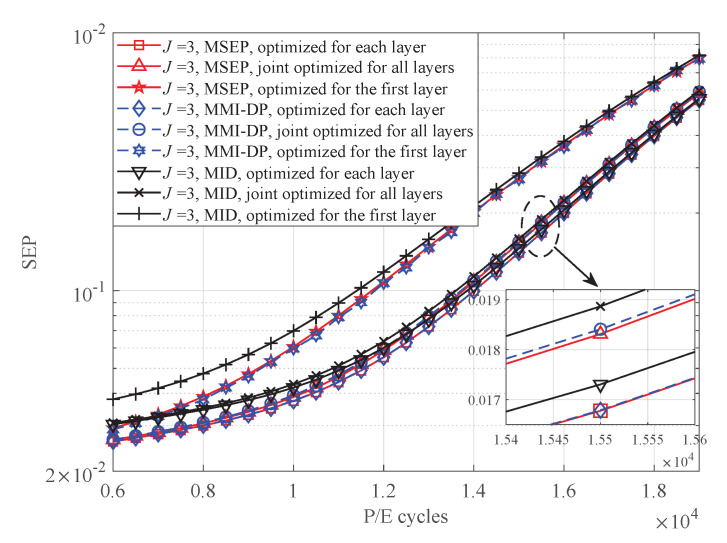
The SEP of the MSEP, MMI-DP, and MID quantizers with 3 read-voltage thresholds over different P/E cycles at $t=5\times 10^{6}$ s.

**Figure 7 entropy-25-00965-f007:**
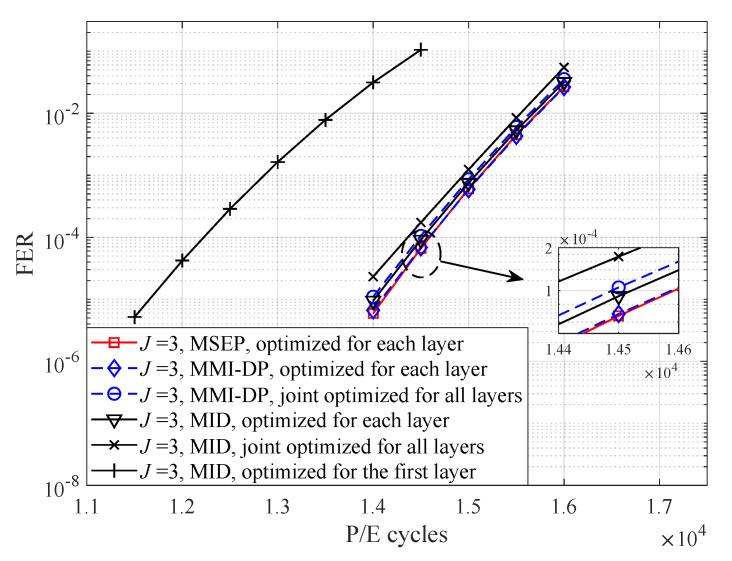
FER performance of LDPC *4K-code* over different P/E cycles with 3 read-voltage thresholds, $I_{\max}$ = 25 and $t=5\times 10^{6}$ s.

**Figure 8 entropy-25-00965-f008:**
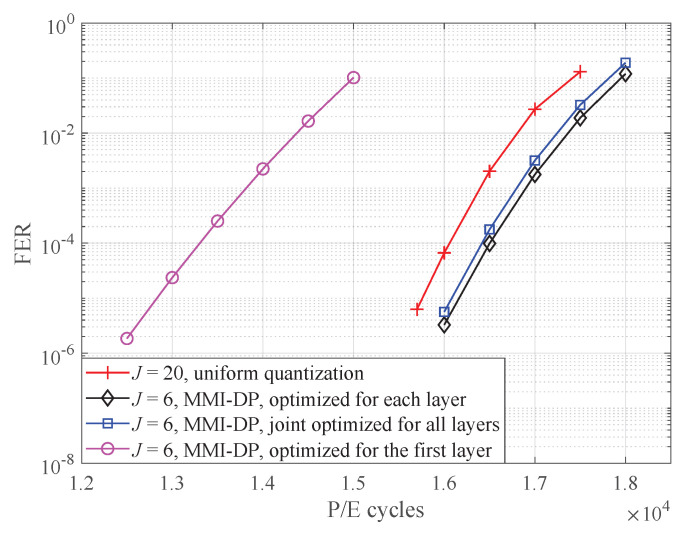
FER performance of LDPC *4K-code* over different P/E cycles with $I_{\max}$ = 25 and $t=5\times 10^{6}$ s.

**Table 1 entropy-25-00965-t001:** Parameters of 3D NAND Flash Memory with *K* layers.

Variable A	Variable A=(α×PE+β)×log(t)+γ×PE+δ				Variable B	Variable B$=\alpha \times k^{3}+\beta \times k^{2}+\gamma \times k$		
	α	β	γ	δ		α	β	γ
$\mu_{s_{0}}^{*}$	$1.01\times 10^{-4}$	0.74	$4.2\times 10^{-3}$	−67.27	$\mu_{l_{0,k}}$	0	−0.028	1.94
$\mu_{s_{1}}^{*}$	$-1.94\times 10^{-5}$	−0.4	$5.14\times 10^{-4}$	106.47	$\mu_{l_{1,k}}$	0	0	0.0075
$\mu_{s_{2}}^{*}$	$-4.71\times 10^{-5}$	−0.7	$1.94\times 10^{-4}$	183.58	$\mu_{l_{2,k}}$	0	0	−0.0447
$\mu_{s_{3}}^{*}$	$-7.37\times 10^{-5}$	−1.2	$4.68\times 10^{-4}$	252.85	$\mu_{l_{3,k}}$	0	0	−0.0308
$\sigma_{s_{0}}^{*}$	$1.2\times 10^{-5}$	−0.1	$2.1\times 10^{-4}$	14.01	$\sigma_{l_{0,k}}$	0	−0.0048	0.185
$\sigma_{s_{1}}^{*}$	$-1.34\times 10^{-6}$	0.0098	$1.56\times 10^{-4}$	8.2	$\sigma_{l_{1,k}}$	0	−0.0045	0.153
$\sigma_{s_{2}}^{*}$	$-2.12\times 10^{-6}$	0.0098	$1.09\times 10^{-4}$	9.65	$\sigma_{l_{2,k}}$	$-1.8\times 10^{-5}$	$9.1\times 10^{-4}$	−0.037
$\sigma_{s_{3}}^{*}$	$2.87\times 10^{-6}$	0.014	$8.5\times 10^{-5}$	9.83	$\sigma_{l_{3,k}}$	$7.86\times 10^{-5}$	−0.0034	0.0129

PE: number of program/erase cycles; *t*: data retention time/second; *k*: layer index.

## Data Availability

The data presented in this study are available on request from the corresponding author.

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
