# Peer review of "Channel Modeling and Quantization Design for 3D NAND Flash Memory"

_entropy, 2023, doi:10.3390/e25070965_

Round 1

Reviewer 1 Report

In general the paper is well written and discusses interesting and significant for  practice problem. The paper can be published, however, a minor improvement can be done.

    Too many abbreviations are used that makes the paper heavy for reading. I recommend their use to be decreased (modern copy paste technology makes this task easy) and a list of abbreviations to be added.

    Sometimes the reader hesitates if a given fact is taken from cited papers or is derived by authors (e.g. (2),(3) and text around). All variables including used i cited results should be carefully defined and explained.

    The simulation with  LDPC codes maybe needs more explanations.

Reviewer 2 Report

This paper proposed channel modeling and quantization design for 3D NAND flash memory.

The reviewer generally supports the acceptance of the manuscript. Some minor issues that need to be clarified in order to improve the quality of the manuscript.

1. It is interesting to analyze whether the temperature also impacts the performance or error rate of 3D flash memory. The reviewer noticed that there have been many works discussing the temperature-aware 3D flash memory (e.g., Yixin Luo et al. [HPCA 2018], Yi Wang et al. [TCAD 2020]). It is not fully convincing that high temperature will lead to more error rates. The reviewer guesses that this paper also has the similar issue, as temperature will definitely impact the results.

2. In terms of process variation, there have been several state-of-art process variation aware techniques to enhance the reliability of 3D flash memory (e.g., Yi Wang et al. [TCAD 2022], [TECS 2017]). These software-hardware codesign techniques could address the process variation issue. The authors should discuss whether the proposed technique could be combined with these existing solutions to further reduce the error rate of flash memory.

3. Another issue is, whether the proposed solution can be applied to other types of 3D NAND flash memory? Whether the proposed solution is a general technique?

The English could be further improved by proof reading.
